# Initiation and Withdrawal of Invasive Ventilation for Patients with Amyotrophic Lateral Sclerosis: A Narrative Literature Review

Reina Ozeki-Hayashi [1], Eisuke Nakazawa [1], Robert Truog [2] and Akira Akabayashi [1,3,*]

1   Department of Biomedical Ethics, Faculty of Medicine, University of Tokyo, Tokyo 113-0033, Japan
2   Harvard Center for Bioethics, Harvard Medical School, Boston, MA 02115, USA
3   Division of Medical Ethics, New York University School of Medicine, New York, NY 10016, USA
*   Correspondence: akira.akabayashi@gmail.com or akirasan-tky@umin.ac.jp; Fax: +81-35841-3319

**Abstract:** Decisions regarding invasive ventilation with tracheostomy (TIV) in patients with amyotrophic lateral sclerosis (ALS) involve serious ethical issues. Cultural differences in the attitudes of patients, caregivers, and physicians toward TIV initiation and withdrawal decisions have been analyzed based on a narrative review approach, comparing the situation between Japan and the U.S. Three main issues were identified regarding the implementation of TIV. The first is the lack of Advance Care Planning. Second, some patients may choose TIV based on the wishes of their physicians or caregivers, even if the patients themselves do not want TIV in the Japanese context. Third is the influence of patient associations, which advocate for the protection of patients' rights. Next, this study identifies the following issues related to TIV discontinuation. The main concern here is cultural differences in legislation and ethical intuitions regarding the discontinuation of TIV. The treatment guidelines for patients with ALS advise physicians to reassure patients that TIV can be withdrawn at any point. However, TIV withdrawal is not explicitly discussed in Japan. Moreover, Japanese ALS treatment guidelines state that ventilation withdrawal is currently impossible, due to a lack of legal support. Most Japanese physicians have told patients that they are not allowed to stop ventilation via such a request. Unlike in the U.S., withholding and withdrawing ventilators are not ethically equivalent in Japan. In conclusion, the decision-making process regarding TIV is difficult, not only for the patients and caregivers, but also for physicians. Even if patients are legally entitled to refuse unwanted treatment, there have been cases in which Japanese physicians have felt an ethical dilemma in stopping TIV for patients with ALS. However, few studies have investigated in detail why physicians oppose the patient's right to discontinue TIV in Japan.

**Keywords:** amyotrophic lateral sclerosis; narrative literature review; invasive ventilation with tracheostomy; ethics; withdrawal

## 1. Introduction

Amyotrophic lateral sclerosis (ALS) is a motor neuron degenerative disease characterized by progressive motor dysfunction that leads to severe disability and respiratory failure. As a treatment for respiratory failure, Non-invasive positive pressure ventilation (hereafter, NPPV) is recommended in the treatment guidelines [1] and effectively relieves patients' symptoms and prolongs life years. However, as the disease progresses, NPPV will be unable to control the symptoms in most cases. Invasive ventilation with tracheostomy (hereafter, TIV) is the only viable option for these patients when fatal respiratory failure becomes imminent.

While TIV can prolong a patient's life, this treatment choice cannot cure the disease itself. As the disease progresses, patients with ALS have to depend on their caregivers for everything in their daily lives, and eventually fall into a so-called "locked-in" syndrome where they cannot communicate with others. For patients, communicating with caregivers

and loved ones is, without a doubt, the greatest concern in life. The transition to instrumental communication, such as the Alphabet Board, should preserve the patient's ability to communicate to the greatest extent possible, prior to the onset of "locked-in" syndrome. With the advancement of research and development, eye-tracking technology [2] has come into wide use, and the detection of intent using a brain computer interface is another area where early practical application is anticipated [3,4]. However, malfunctions in communication always threaten the quality of life of patients. Thus, patients with ALS must make some tough decisions: to go to their glory without TIV, or to survive with TIV, even if it means that they cannot speak with others. Some researchers have reported that all patients with TIV became TLS due to the low rate of TIV in Europe and the U.S. [5,6], while in Japan, 13% of patients with TIV had TLS [7].

Since the TIV decision-making process is so complicated, it is essential for ALS patients to engage in sufficient appropriate discussions in advance with family, friends, healthcare professionals, and caregivers. At that time, physicians' explanations and values have been shown to impact on the decision-making process significantly [8]. Therefore, this literature review will summarize physicians' explanations and values, as well as ALS patients' attitudes toward TIV initiation and withdrawal decisions, mainly in the Japanese and the U.S. context, to begin with. These two countries were chosen because (1): Japan has a high TIV use rate (27–45%), while the U.S. shows a lower TIV use rate (1–14%) [1]; (2) These countries both have advanced economies; (3) They have entirely different medical supply laws and systems; and (4) They have very different Eastern and Western cultural backgrounds. (5) Although European countries would also be suitable for research, the European Union consists of many countries. It is therefore easier to begin with the U.S., where many comparative studies have been conducted outside the field of medical neurology, including some that compare the cultural backgrounds of Japan and the U.S.

## 2. Methods

We conducted a narrative literature review [9]. Firstly, we conducted a rough search using keywords such as amyotrophic lateral sclerosis, ventilation, and withdrawal. Secondly, we identified specific keywords related to our review aim (e.g., amyotrophic lateral sclerosis respiration, artificial, withholding and withdrawing treatment, ethics, terminal care, Japan, etc.), found 75 articles on PubMed, and added a few articles by hand-searching. Thirdly, we reviewed those abstracts and papers. Finally, we summarized and synthesized all the findings and integrated them.

A narrative literature review is an old style of review, presenting a summary and analysis of the available literature on a particular topic of interest [10]. There are no formal guidelines around it, as the approach is nonsystematic [10,11]. It can address multiple questions [11,12], and is well suited to addressing overarching topics [13].

Specific methods of narrative literature review include the following: (1) topic selection, which involves narrowing down the field to topics related to clinical practice and policy, preferably focused on a specific issue; (2) literature search and retrieval, which are related to identifying the literature that is most relevant to the selected topic; (3) critical examination of the literature, which involves identifying methodological issues and knowledge gaps in the reviewed studies and establishing new findings; and (4) finding a logical structure for the narrative literature review and writing it clearly and effectively [10]. An advantage of this method is that it can offer suggestions for understanding the diversity and complexity of an academic research topic [14]. Additionally, it is well suited for extracting general arguments, evaluating prior research, and determining gaps in the current knowledge [11]. It is particularly suitable for presenting philosophical perspectives in a balanced manner, and it is capable of stimulating scholarly discussion among readers to provoke thought and debate [15]. Its main weaknesses include the problem of bias due to the subjectivity involved in the selection of articles [11,12] and the possibility of selecting only literature that supports the author's regulatory opinion, giving too much credence to the preferred hypothesis [11]. To prevent this bias, researchers should refer to systematic reviews to

define the criteria for the adoption and exclusion of search terms, work with trusted coauthors rather than carrying out the research alone [11], clarify selection criteria and interpretations whenever possible, and clearly define the scope of the review. Examples of narrative literature reviews include Bower (2005) and Cullen (2020) [16,17]. Accordingly, one of the main purposes of this communication is to let researchers worldwide know of this very fast and useful way of summarizing regarding specific topics, taking ALS as an example.

## 3. Overview of TIV for Patients with ALS

The rate of TIV use in patients with ALS varies widely from country to country. For example, it is almost 0% in the U.K., 1–14% in the U.S., 3% in Germany, 11% in Northern Italy, and 27–45% in Japan [1]. Some researchers have suggested that this difference is due to patients' values, socio-cultural factors, medical care systems, and laws. Moreover, as Rabkin et al. have shown, many physicians know that their explanations and values influence their patients' TIV decisions [18].

Patients develop respiratory failure due to progressive paralysis of the respiratory muscles and ball paralysis. In cases of mild respiratory failure, NPPV is the treatment option that improves ALS's prognosis [19]. NPPV allows for immediate initiation of therapy, the ability to eat and speak, and relatively little chance of infection. However, as respiratory failure progresses further, NPPV with low ventilation efficiency is unlikely to improve symptoms [20]. For example, if patients must use NPPV for more than 12 h per day, if the Forced Vital Capacity (FVC) is below 50%, or if respiratory symptoms are severe, we should consider the initiation of TIV [21]. TIV is invasive and requires time and effort to start. In addition, it has the disadvantages of requiring periodic tube changes, increased expectoration due to the placement, and increased risk of bleeding, pain, and infection. However, it has the advantage of superior ventilation efficiency and easier phlegm suctioning, compared to NPPV [20].

The treatment guidelines for patients with ALS advise physicians to reassure patients that TIV can be withdrawn at any point [22]. However, withdrawal of TIV is not explicitly discussed in Japan. In general, the right to refuse unwanted treatment is regarded as an inherent right of patients in Japan as well. The Lisbon Declaration on the Rights of the Patients [23], which clearly states that patients have the right to refuse unwanted treatment, is well known in medical institutions, and is often posted on hospital websites. However, it is not applied to the specific withdrawal of TIV in patients with ALS. Japanese ALS treatment guidelines also state that the withdrawal of ventilation is currently impossible due to a lack of legal support [24]. Moreover, unlike in the U.S., withholding and withdrawing ventilators are not ethically equivalent in Japan [25]. In the U.S., the predominant view seems to be that withholding and withdrawing ventilators are ethically equivalent in a consequentialist manner. By contrast, in Japan, there is a tendency to emphasize the distinction between physicians' acts and omissions, and to avoid the negative effects of their actions.

This can lead to cases where patients with ALS who wish to discontinue treatment after careful consideration suffer because their wishes are not respected. In an extreme case, a few years ago, a patient with ALS who had a strong desire to withdraw TIV asked his/her doctors via a social networking service to administer a lethal drug without a careful decision-making process [26]. Furthermore, 42 percent of patients with ALS have undergone emergency TIV due to rapid respiratory failure. In most of these cases, TIV was administered without adequate discussion [18].

## 4. Issues Related to TIV Initiation
### 4.1. The Attitudes of Patients with ALS towards TIV Initiation

We demonstrate the result of surveys of ALS patients' attitudes toward TIV. In Borasio et al.'s study for patients with ALS in Japan [8], it was found that 21% of patients started TIV upon their doctors' recommendations, 10% by their own choice, and 12% upon the wishes

of family members. By contrast, 42% of patients started TIV in an emergency without prior discussion. In addition, 35% of patients rejected TIV based on their own personal wishes. For 25% of patients, their family members chose not to use it. The reason for why these patients did not start TIV was not a "financial burden," but rather their fear of ending up in "locked-in syndrome." It is noteworthy that patients knew that TIV treatments were entirely covered by the public medical insurance system in Japan.

In a comparative study of patients with ALS in the U.S. and Japan, only about 20% of patients with advanced respiratory symptoms favored TIV in both countries [27]. The survey suggests that the attitudes of physicians and caregivers, and the lack of Advance Care Planning (hereafter, ACP) may have influenced the patients' decision to start TIV [27].

### 4.2. Caregivers' Attitudes toward TIV Initiation

We discuss a comparative survey for ALS caregivers between Japan and the U.S. [28]. In this study, caregivers who wanted patients to undergo TIV accounted for 33% in the U.S. and 53% in Japan, respectively. The rate of concordance between the TIV preferences of caregivers and patients was higher in the U.S. than Japan. Based on their personal experiences, Ogino and Babayev (2020) show that many caregivers do not want TIV if they have the same disease, but often want it for their family members [29]. Although the study notes that these family wishes may seem selfish, the authors surmise that ALS patients with TIV in Japan may respect their families' wishes. Patients should live in a way that helps their families. By doing so, they can achieve happiness for themselves as well. Although this personal belief may be one reason for why patients with ALS favor TIV, the present research results suggest that some Japanese patients may choose TIV based on their caregivers' wishes, even if they themselves do not want TIV. We surmise that this is less likely to happen in the more individualistic U.S. society.

### 4.3. Physicians' Attitudes towards TIV Initiation

A study reported that 79% of neurologists in the U.S. and 36% in Japan said they rarely recommend TIV to ALS patients [18]. In their discussions of TIV with ALS patients, 70% of U.S. physicians said "[My] role is to present treatment options along with my recommendation." In comparison, 60% of Japanese physicians said "[I] present the options and let the patient decide (without the doctor's recommendation)." When asked what they thought about the reasons for patients not receiving TIV, U.S. neurologists had several answers, including "they are disabled and ready to die," "financial burdens of care," and "they have to enter institutions if they choose TIV and do not want to enroll." On the other hand, Japanese neurologists answered, "the reason the patients don't choose TIV is that we cannot stop TIV once it starts," and "the patient does not want to live any longer." When asked if they would like to undergo TIV themselves if they had ALS, 76% of U.S. physicians and 72% of Japanese physicians answered that they would not want to undergo TIV themselves. According to the discussion section of this paper, ALS patient advocacy organizations in the U.S. have not made any statements in support of TIV. In contrary, in Japan, such associations have advocated for TIV use [18]. Therefore, this paper infers that Japanese physicians who recommend TIV treatments to their patients probably echo the statements of Japanese ALS associations, which advocate for the protection of patients' rights (contrary to their preferences). Some researchers have argued that Japanese doctors are influenced by the ideas of the ALS advocacy group but are trying to balance both approaches [29].

### 5. Issues Related to the Discontinuation of TIV

### 5.1. A Patient's Request to Withdraw TIV

We will show the physician's experience of a patient's request to withdraw TIV. In a comparative survey of physicians in the U.S. and Japan [18], 71% of physicians in the U.S. and 8% in Japan said that they had asked their patients about stopping TIV. Additionally, 71% of U.S. physicians and 49% Japanese said they had received a patient's request to

withdraw TIV. Compared to the 2010 study in Japan, the number of physicians who have received discontinuation requests from patients has increased slightly (21%) [30]. In both surveys, most Japanese physicians told their patients that they were not permitted to stop ventilation in response to such a request. This is because, as already mentioned, Japanese law does not clearly stipulate the withdrawal of treatment, leaving it ambiguous as to whether physicians are exempt from liability for the withdrawal of treatment that results in euthanasia.

### 5.2. Does the Patient Have the Right to Discontinue TIV?

We explore physicians' attitudes toward patients requesting TIV withdrawal.

In a survey of the attitudes of 55 physicians in five European countries towards hypothetical patients with the psychological and physical suffering associated with ALS [31], 73% suggested proactively stopping TIV due to psychological distress, and 92% said they would stop TIV on request. In contrast, 44% suggested stopping TIV due to physical distress, and 83% of physicians would stop TIV. This indicates that the physicians in Western countries have a relatively positive view of stopping TIV in patients with ALS who are suffering from psychological distress.

In a qualitative study examining the ethical issues involved in stopping TIV for patients with motor neuron intractable diseases (including ALS), almost all of the 24 participating British physicians [32] said that patients had a legal right to stop unwanted treatment (including TIV). They also acknowledged that some patients sincerely wanted to discontinue TIV because they did not want to continue living any longer and did not want to lose any more abilities. In this study, even among physicians who believed that patients have the right to stop unwanted TIV, some felt emotionally uncomfortable with stopping TIV for patients with ALS. For example, some physicians felt emotionally burdened by ALS patients who die soon after TIV withdrawal, because TIV allowed them to communicate and think clearly. Some physicians felt that TIV withdrawal resembles "physician-assisted suicide" because patients wanted to end their lives. Additionally, even when the patients had legalized advance directives, physicians sometimes found it difficult to respect the wishes of unconscious patients with ALS to discontinue TIV.

### 5.3. Should Patients Be Granted the Right to Discontinue TIV?

Finally, we discuss the physicians' attitudes toward the patients' right to discontinue TIV.

In a national attitude survey of neurologists in Japan [30], 24% of physicians said they should not allow the right to withdraw TIV for patients with ALS. In comparison, 59% said they should allow such a right if the patients' decisions meet some requirements, such as if all family members agree, or if advance directives are provided.

In another study in 2011 [33], researchers surveyed physicians using a hypothetical case in which all stakeholders agreed with an ALS patient's wish to discontinue TIV. While 67% of physicians said they would stop TIV if they were asked a general opinion on this case, 63% answered that they would not withdraw TIV if they were the attending physicians for this patient. The results suggest that there is a gap between generic theories about the treatment of patients with ALS, and specific, individualized recommendations. In this survey, the physicians commented that they should not withdraw TIV because "a new treatment may be available" or "the doctors should respect the patient's dignity." Contrastingly, some physicians in favor of stopping TIV commented, "Even though there is a little expectation for improvement or cure in the future, isn't it more ethical to recognize the right not to want to live with pain and suffering for an uncertain period until then?".

In most Japanese arguments about the patient's right to withdraw TIV (most researchers are not medical professionals), commentators opposed patient rights. Firstly, patients' decisions are unreliable because patients may change their minds during the disease process [34]. Secondly, "[We] cannot argue for the right to discontinue TIV unless we can create a society where all patients with ALS can receive sufficient qualitative and

quantitative care. In other words, they insist that if we allow the right to discontinue TIV, there are huge concerns that patients with ALS with TIV may be at risk of silent social pressure to compel them to stop their ventilation (slippery slope theory)" [34,35].

Medical professionals and patients involved in such decision-making often face profound ethical dilemmas. It is desirable to establish a system that allows multiple medical providers to collaborate in deciding how to deal with such issues, and if possible, to establish a clinical ethics consultation system. Even after a patient, the patient's family, and the medical team have decided, it is desirable to develop a system to monitor the decision's progress and to consider whether any changes occur in the patient's wishes.

## 6. Conclusions

In this literature review on the initiation and discontinuation of TIV in patients with ALS, we focused mainly on physicians' attitudes. We found that the decision-making process regarding TIV was difficult, not only for the patients and their families, but also for the medical professionals involved. Even if patients were legally entitled to refuse unwanted treatment, there were cases in which physicians felt an ethical dilemma in stopping TIV for patients with ALS. Very little research has been conducted to investigate in detail why physicians oppose the patient's right to discontinue TIV in Japan. Moreover, we could not find any empirical research on why Japanese physicians recommend TIV treatment for their patients, contrary to their own preference. Why do they engage in such inconsistent behavior? Is there any ethical or emotional conflict involved for such doctors? Previous studies have failed to answer these questions. How do Japanese physicians deal with the discrepancy between their preference for TIV and their practical recommendations for patients with ALS? We would like to support patients, families, and medical professionals involved in these challenging decision-making processes. We need further research to investigate the background factors in these challenging decision-making processes.

**Author Contributions:** Conceptualization, R.O.-H. and R.T.; writing—original draft preparation, R.O.-H. and R.T.; writing—review and editing, R.O.-H., R.T., E.N. and A.A.; project administration, A.A. All authors have read and agreed to the published version of the manuscript.

**Funding:** This research received no external funding.

**Institutional Review Board Statement:** Not applicable.

**Informed Consent Statement:** Not applicable.

**Data Availability Statement:** Not applicable.

**Acknowledgments:** The authors would like to thank Lisa Moses of the Harvard University for her invaluable assistance on this manuscript.

**Conflicts of Interest:** The authors declare no conflict of interest.

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
