# Peer review of "Initiation and Withdrawal of Invasive Ventilation for Patients with Amyotrophic Lateral Sclerosis: A Narrative Literature Review"

_2571-8800, doi:10.3390/j5030027_

Round 1

Reviewer 1 Report

The article touches on an important topic of rationale, initiation, and possibility of withdrawal of invasive ventilation in this very particular group of neuromuscular patients. I find this study clinically relevant and well written, and advocate publication.

I have only a few minor critical comments on the manuscript:

1. the title may be shortened, home invasive ventilation is always via a tracheostomy and there is no need to say it.

2. Introduction is very long and its second part refers more to the methods used for the review. I propose to create a new chapter "Methods" starting with line 65, page 2.

3. Chapter 2. Overview... tells more about withdrawal than general information on tracheostomy in ALS. I propose to describe the indications for and timing of tracheostomy in ALS and the differences in the care which have to be provided to tracheostomized versus NIV patients.

4.  Authors suggest that the worst scenario in ALS patients is a locked-in syndrome.  Are there any data on the frequency of this syndrome? Considering my practice usually patients die due to complications before this syndrome occurs. Moreover, the authors should mention modern possibilities of communication, like writing with eyes with the use of special software. 

5. There is no need to use words: firstly, next, and thirdly, if there are separate paragrafs divided by numbers.

6. Authors should explain the legal regulation concerning euthanasia in Japan and USA because discontinuation of ventilation in ALS would be one of the forms of euthanasia. Without this explanation such sentence as: "most Japanese physicians told their patients that they were not permitted to stop ventilation in response to such a request." will not be understandable.

Author Response

We sincerely thank you for your careful review of our paper. We also received six very useful suggestions. All six suggestions are appropriate and very useful in improving our paper.

  1. the title may be shortened, home invasive ventilation is always via a tracheostomy and there is no need to say it.

Thank you. We have shortened the title as follows

Initiation and withdrawal of invasive ventilation for patients with amyotrophic lateral sclerosis: A narrative literature review

  1. Introduction is very long and its second part refers more to the methods used for the review. I propose to create a new chapter "Methods" starting with line 65, page 2.

Thank you. We have added a section on 2. Methods in accordance with your suggestion.

  1. Chapter 2. Overview... tells more about withdrawal than general information on tracheostomy in ALS. I propose to describe the indications for and timing of tracheostomy in ALS and the differences in the care which have to be provided to tracheostomized versus NIV patients.

Thank you. W have added the following as you suggested.

Patients develop respiratory failure due to progressive paralysis of the respiratory muscles and ball paralysis. In cases of mild respiratory failure, NPPV is the treatment option that improves ALS’s prognosis [19]. NPPV allows for immediate initiation of therapy, the ability to eat and speak, and relatively little chance of infection. However, as respiratory failure progresses further, NPPV with low ventilation efficiency is unlikely to improve symptoms [20]. For example, if patients must use NPPV for more than 12 hours per day, if the Forced Vital Capacity (FVC) is below 50%, or if respiratory symptoms are severe, we should consider the initiation of TIV [21]. TIV is invasive and requires time and effort to start. In addition, it has the disadvantages of requiring periodic tube changes, increased expectoration due to the placement, and increased risk of bleeding, pain, and infection. However, it has the advantage of superior ventilation efficiency and easier phlegm suctioning, compared to NPPV [20].

  1. Bourke SC, Tomlinson M, Williams TL, et al. Effects of non-invasive ventilation on survival and quality of life in patients with amyotrophic lateral sclerosis: a randomized controlled trial. Lancet Neurol. 2006; 5:140-147.
  2. Practical Guideline for Amyotrophic Lateral Sclerosis (ALS) 2013, Societas Neurologica Japonica, Nankodo., Ltd., Tokyo (in Japanese)
  3. Gruis KL, Lechtzin N. Respiratory therapies for amyotrophic lateral sclerosis: a primer. Muscle Nerve. 2012; 46: 313-331.

  1. Authors suggest that the worst scenario in ALS patients is a locked-in syndrome. Are there any data on the frequency of this syndrome? Considering my practice usually patients die due to complications before this syndrome occurs. Moreover, the authors should mention modern possibilities of communication, like writing with eyes with the use of special software.

Thank you. In Europe and the US, it is reported that all patients with TIV have TLS, perhaps because of the low rate of TIV fitting, but a report from Japan showed that 13% of patients with TIV had TLS. We add the following sentence.

For patients, communicating with caregivers and loved ones is, without a doubt, the greatest concern in life. The transition to instrumental communication, such as the Al-phabet Board, should preserve the patient's ability to communicate to the greatest extent possible, prior to the onset of “locked-in” syndrome. With the advancement of research and development, eye-tracking technology [2] has come into wide use, and the detection of intent using a brain computer interface is another area where early practical application is anticipated [3, 4]. However, malfunctions in communication always threaten the quality of life of patients.

Some researchers have reported that all patients with TIV became TLS due to the low rate of TIV in Europe and the U.S. [5, 6], while in Japan, 13% of patients with TIV had TLS [7].

  1. Polkey MI, Lyall RA, Moxham J, et al: Respiratory aspects of neurological disease. J Neurol Neurosurg Psychiatry 1999; 66: 5―15
  2. Borasio GD, Voltz R: Palliative care in amyotrophic lateral sclerosis. J Neurol 1997; 244 Supple 4: S11―S17
  3. Kawata A, Mizoguchi K, Hayashi H: A nationwide survey of ALS patients on tracheostomy positive pressure ventilation (TPPV) who developed a totally locked-in state (TLS) in Japan. Clin Neurol, 48: 476―480, 2008 (in Japanese)
  4. Caligari M, Godi M, Guglielmetti S, Franchignoni F, Nardone A. Eye tracking communication devices in amyotrophic lateral sclerosis: impact on disability and quality of life. Amyotroph Lateral Scler Frontotemporal Degener 2013, 14(7-8), 546-552. doi:10.3109/21678421.2013.803576
  5. Pasqualotto E, Matuz T, Federici S, et al. Usability and Workload of Access Technology for People With Severe Motor Impairment: A Comparison of Brain-Computer Interfacing and Eye Tracking. Neurorehabil Neural Repair 2015, 29(10), 950-957. doi:10.1177/1545968315575611
  6. Pugliese R, Sala R, Regondi S, Beltrami B, Lunetta C. Emerging technologies for management of patients with amyotrophic lateral sclerosis: from telehealth to assistive robotics and neural interfaces. J Neurol 2022, 269(6), 2910-2921. doi:10.1007/s00415-022-10971-w

  1. There is no need to use words: firstly, next, and thirdly, if there are separate paragraphs divided by numbers.

Thank you very much. We will simplify the expression.

  1. Authors should explain the legal regulation concerning euthanasia in Japan and USA because discontinuation of ventilation in ALS would be one of the forms of euthanasia. Without this explanation such sentence as: "most Japanese physicians told their patients that they were not permitted to stop ventilation in response to such a request." will not be understandable.

Thank you very much. We have already explained the lack of legal support for treatment discontinuation in Japan in Lines 107-120. In addition, we would like to add the following additional explanation to Line 188 for clarification and context.

This is because, as already mentioned, Japanese law does not clearly stipulate the withdrawal of treatment, leaving it ambiguous as to whether physicians are exempt from liability for the withdrawal of treatment that results in euthanasia.

Reviewer 2 Report

The manuscript consists of total 7 pages, including the list of 26 literature sources. The narrative review of the ethical and legal problems associated with the use of invasive ventilation in patients with progressing motor neuron dysfunction ALS finally preventing them from being able to breathe by their own while their consciousness stays for much longer time still intact. Discussing the ethical aspects of modern medicine capabilities use is essential for finding the balance between what can be used and what shall be used - in the best interest of the particular patient's good - so it is highly valuable that the Authors decided to analyze the topic. The manuscript is written in good quality English and presents as an interesting and engaging read, fitting into the scope of works published by the Journal. The title of the article is adequate to its contents. The Abstract mirrors the contents of the main text.

The article has a logical structure, discussing first the essential facts concerning both the neurological pathology ALS and its ventilation treatment, then the issues related to starting and discontinuing the treatment, contrasting the different realities in the US and Japan. However, in the end portion of the article, as well as in its conclusions, I miss some discussion of the narrative review results. The Authors need to express some own, original opinions about the topic they raised, or maybe point at some available solutions they can see for the identified problems and some directions of actions to be taken, as well as some practical hints for both patients and physicians who confront the difficult ethical and legal dilemmas associated with using the discussed TIV therapy in the ALS patients. Adding such a part to the article and its conclusions would significantly increase its scientific value as another voice contributing to the discussion of the practice-relevant topic, that would be not purely reporting but also possibly creating some new solution idea, or at least pointing towards some existing one based on the analyzed and presented data.

The literature references are relevant to the topic and in the vast majority recent enough. However, the number of sources finally referred to by the Authors seems a bit small for a review type article.

Author Response

We are very encouraged by your comments, and we sincerely respect the fact that Reviewer2 has taken the time to read our manuscript. Thank you very much.

Reviewer2 also provided the following very helpful comments. We agree with his points and have added to them.

However, in the end portion of the article, as well as in its conclusions, I miss some discussion of the narrative review results. The Authors need to express some own, original opinions about the topic they raised, or maybe point at some available solutions they can see for the identified problems and some directions of actions to be taken, as well as some practical hints for both patients and physicians who confront the difficult ethical and legal dilemmas associated with using the discussed TIV therapy in the ALS patients.

Our additions are as follows:

Medical professionals and patients involved in such decision-making often face pro-found ethical dilemmas. It is desirable to establish a system that allows multiple medical providers to collaborate in deciding how to deal with such issues, and if possible, to estab-lish a clinical ethics consultation system. Even after a patient, the patient's family, and the medical team have decided, it is desirable to develop a system to monitor the decision's progress and to consider whether any changes occur in the patient’s wishes.
